# Potential Effect of *Pseudevernia furfuracea* (L.) Zopf Extract and Metabolite Physodic Acid on Tumour Microenvironment Modulation in MCF-10A Cells

**DOI:** 10.3390/biom11030420

**Published:** 2021-03-12

**Authors:** Klaudia Petrova, Martin Kello, Tomas Kuruc, Miriam Backorova, Eva Petrovova, Maria Vilkova, Michal Goga, Dajana Rucova, Martin Backor, Jan Mojzis

**Affiliations:** 1Department of Pharmacology, Faculty of Medicine, Pavol Jozef Šafárik University, 040 01 Košice, Slovakia; claudy.petrova@gmail.com (K.P); tomaskuruc@gmail.com (T.K.); 2Department of Pharmaceutical Technology, Pharmacognosy and Botany, University of Veterinary Medicine and Pharmacy, 041 81 Košice, Slovakia; miriambackorova@gmail.com; 3Department of Anatomy, Histology and Physiology, University of Veterinary Medicine and Pharmacy, 041 81 Košice, Slovakia; petrovova.e@gmail.com; 4Department of NMR Spectroscopy, Institute of Chemistry, Faculty of Science, Pavol Jozef Šafárik University, Moyzesova 11, 040 11 Košice, Slovakia; maria.vilkova@upjs.sk; 5Core Facility Cell Imaging and Ultrastructure Research, University of Vienna, Althanstrasse 14, 1090 Vienna, Austria; michal.goga@univie.ac.at; 6Department of Botany, Institute of Biology and Ecology, Faculty of Science, Pavol Jozef Šafárik University, Mánesova 23, 041 67 Košice, Slovakia; dajana.rucova@gmail.com (D.R.); martin.backor@upjs.sk (M.B.)

**Keywords:** tumour microenvironment, EMT, MCF-10A, fibroblasts, HUVECs, angiogenesis, lichens, secondary metabolites

## Abstract

Lichens comprise a number of unique secondary metabolites with remarkable biological activities and have become an interesting research topic for cancer therapy. However, only a few of these metabolites have been assessed for their effectiveness against various in vitro models. Therefore, the aim of the present study was to assess the effect of extract *Pseudevernia furfuracea* (L.) Zopf (PSE) and its metabolite physodic acid (Phy) on tumour microenvironment (TME) modulation, focusing on epithelial–mesenchymal transition (EMT), cancer-associated fibroblasts (CAFs) transformation and angiogenesis. Here, we demonstrate, by using flow cytometry, Western blot and immunofluorescence microscopy, that tested compounds inhibited the EMT process in MCF-10A breast cells through decreasing the level of different mesenchymal markers in a time- and dose-dependent manner. By the same mechanisms, PSE and Phy suppressed the function of Transforming growth factor beta (TGF-β)-stimulated fibroblasts. Moreover, PSE and Phy resulted in a decreasing level of the TGF-β canonical pathway Smad2/3, which is essential for tumour growth. Furthermore, PSE and Phy inhibited angiogenesis *ex ovo* in a quail embryo chorioallantoic model, which indicates their potential anti-angiogenic activity. These results also provided the first evidence of the modulation of TME by these substances.

## 1. Introduction

Breast cancer (BC) is a multifaceted global health issue and one of the most common reasons for cancer-related mortality in women worldwide [1,2]. According to GLOBOCAN 2020, about 2,261,419 women were estimated to be newly diagnosed with breast cancer, which makes up 24.5% of all incidences of cancer among women. Despite many therapeutic approaches for BC patients, such as chemotherapy, radiation and/or surgery, the prevalence of BC has still increased [1,3,4]. Therefore, new therapeutic strategies are required to be investigated. In the past decades, many investigators have focused attention primarily on tumour cells. But recent findings suggest that tumour progression does not only depend on tumour cells themselves, and the changes in the tumour microenvironment (the tumour stroma, surrounding blood vessels and immune cells) have also been associated as the critical regulators in tumour development and progression [5,6]. In particular, there is evidence that also the immune system can influence and promote tumour development, increasing cancer heterogeneity, and be the source of biomarkers for potential research and diagnostics [7,8]. Thus, understanding the regulatory mechanisms affecting the tumour microenvironment (TME) can reveal new therapeutic options in cancer research.

The TME consists of extracellular matrix (ECM) and several heterotypic cells, such as cancer-associated fibroblasts, immune and endothelial cells [9,10]. A variety of factors secreted by these cells can lead to epithelial–mesenchymal transition (EMT), which is an initial step of the metastatic cascade [11,12,13,14]. Because metastasis appears as a prominent problem in cancer therapy, EMT represents the main target of our research as well as various components of TME, such as cancer associated fibroblasts (CAFs) and process of angiogenesis. EMT is a multistep reversible process which is responsible for morphological and behavioural changes in cells. Through EMT, epithelial cells lose their intracellular adhesion and cell polarity to gain a motile mesenchymal phenotype [15]. Metastatic cells are characterised by increased motility, which allows them to escape the primary tumour and colonise distant organs [16,17]. Under the influence of various cytokines and growth factors, epithelial cells change their shapes from cubic to spindle-shaped, showing decreased expression of epithelial markers (E-cadherin, β-catenin) and increased expression of mesenchymal markers (N-cadherin, fibronectin, vimentin) [18,19]. Circulating tumour cells (CTCs) found in the bloodstream, that are shed both by the primary and metastatic sites, represent rare but important part of TME. CTCs in breast cancer development can potentially predict poorer response to conventional anticancer therapy, as reviewed in Reference [20]. Existing evidence suggests that a high percentage of ER (oestrogen receptor)-negative CTCs or discordance in HER-2 (human epidermal growth factor receptor 2) status influence the possible escape mechanism to endocrine therapy and response to anticancer treatment [21,22].

Among the plenty of signalling pathways, growth factors and cytokines that induce the EMT in cancer, transforming growth factor (TGF-β) has been shown to play a pivotal role [23]. The TGF-β signalling pathway is a key player involved in cell proliferation, differentiation, migration and apoptosis [24,25]. TGF-β induces EMT through a Smad-dependent and a Smad-independent pathway [26]. The Smad-dependent signalling pathway begins with the binding of TGF-β to tyrosine kinase receptors (TβR-I, TβR-II). Binding of TGF-β to TβR-II allows activation of TβR-I and leads to induction of the Smad2 and Smad3 signalling pathways. Phosphorylated Smads then form an oligomeric complex with Smad4, which subsequently translocates to the nucleus to control transcription of target genes through their interaction with various transcription factors, such as Snail, Slug, ZEB1, ZEB2 and Twist [27,28,29]. In addition to the Smad-dependent signalling pathway, TGF-β can also activate non-Smad pathways, such as PI3K/AKT, MAPK, ERK1/2, RhoA and Ras [11].

TGF-β is also responsible for trans-differentiation of noncancerous fibroblasts into CAFs [30], which are one of the largest populations found in TME of breast cancer stroma [31]. Different sources of CAF contribute to the heterogeneity of these cells and to the difficulty of distinguishing them from others that are present in TME. Therefore, there are several markers that identify CAFs, such as alpha-smooth muscle actin (α-SMA), fibroblast-specific protein-1 (FSP-1, also called S100A4), platelet-derived factor receptor (PDGFR), fibroblast activating protein (FAP), podoplanin (PDPN), tenascin-C (TNC), vimentin and neuronal glial antigen 2 (NG-2). However, it is important to note that none of these markers are strictly specific only for cancer-associated fibroblasts and that other cancer or stromal cells may exhibit some of these markers [32].

Finally, a network of dilated and heterogeneous blood vessels contributes to the development of the TME as well [33]. Angiogenesis, the formation of new blood vessels from pre-existing ones, is one of the major characteristics of the TME [34]. Among many others, vascular endothelial growth factor (VEGF) is one of the main key regulators of angiogenesis, that is secreted not only by cancer cells but also by stromal cells [35]. Anti-angiogenic therapy is thought to be an important therapeutic approach for cancer research.

Increasing attention is currently focused to natural products with potential antitumour effects. One of natural substances with potent inhibitory activity on tumour cells are lichens. Lichens are ubiquitous symbionts of fungi (mycobiont) and algae (photobiont) and/or cyanobacteria (cyanobiont) [36]. They are characterised by the production of more than 1000 secondary metabolites with significant pharmacological effects, including antitumour activity [37]. One of these has been tested in the current work.

The aim of the present study is to assess the potential effect of lichen extract *P. furfuracea* (L.) Zopf and metabolite physodic acid on tumour microenvironment modulation in normal human mammary epithelial cells as a model system. This study focused primarily on epithelial–mesenchymal transition in two different types of normal cell lines (breast MCF-10A, fibroblasts BJ-5ta). Moreover, we wanted to estimate a time- and a dose-response of the tested substances. Lastly, the potential anti-angiogenic effect of PSE and Phy was tested using the *ex ovo* CAM assay.

## 2. Material and Methods

### 2.1. Lichen Material and Isolation of Tested Compounds

*Pseudevernia furfuracea* (L.) Zopf was collected from barks of *Picea abies*, at Kojšovská hoľa (48.781049, 20.978567) in Volovské vrchy (Košice, Slovakia) during September 2019. *P. furfuracea* (L.) Zopf was collected and determined by Dr. Goga. The lichen specimen was deposited in herbarium of P.J. Šafárik in Košice (KO35800). Lichen extract *P. furfuracea* (L.) Zopf contains, as major compounds in the cortex, atranorin, chloratranorin and physodic acid, as a medullar major compound [38].

The lichen thalli were rinsed with distilled water to get rid of particles which do not belong to the lichen and air-dried at room temperature (26 °C). Ten grams (dry weight) of lichen thalli were put into a glass beaker and rinsed by 300 mL of acetone for extraction of secondary metabolites according to Solhaug and Gauslaa [39]. The lichen material was mixed with a magnetic stirrer for 24 h. The supernatant was evaporated by a rotary evaporator and extract of secondary metabolites were stored for further experiments. One mg of dry extract was solved in acetone and TLC (Thin Layer Chromatography) plate identification of lichen substances was performed. The ratio of mobile phase for separation of lichen compounds by column chromatography was 3:7:0.4 (etylacetate:cyclohexane:acetic acid). Collected fractions with the same metabolite were put into the evaporating flask and liquid phase was evaporated again. Finally, the five fractions were isolated by column chromatography and used for further identification by High-Performance Liquid Chromatography (HPLC) and Nuclear Magnetic Spectroscopy (NMR).

### 2.2. High-Performance Liquid Chromatography (HLPC)

Extract and all fractions were performed by the semi-preparative method HPLC. 1 mg/2 mL of acetone extract and all fractions were analysed by gradient [40] under the following conditions: A 7 μm column Kromasil SGX C_18_, flow rate 0.7 mL × min^−1^, mobile phase: A = H_2_O:Acetonitrile:H_3_PO_4_ (80:19:1) and B = 90% acetonitrile, gradient program: 0 min 25% B, 5 min 50% B, 20 min 100% B, 25 min 25% B. Detection was performed at a wavelength of 254 nm (detector Ecom LCD 2084; Ecom, Prague, Czech republic). Atranorin, chloroatranorin, 3-hydroxyphysodic acid, physodalic acid and physodic acid were used as standards (internal database of the Department of Botany, University of Pavol Jozef Šafárik in Košice).

### 2.3. Nuclear Magnetic Resonance (NMR) Spectroscopy

NMR spectra were recorded on a VNMRS spectrometer (Varian) operating at 599.87 MHz for ^1^H and 150.84 MHz for ^13^C at 299.15 K. Chemical shifts (*δ* in ppm) are given from internal solvent, CD_3_OD-d_4_ (3.31 ppm for ^1^H and 49.0 ppm for ^13^C).

### 2.4. Cell Culture

The MCF-10A (human mammary gland) cell line was purchased from American Type Culture Collection (ATCC) and cultured in a medium consisting of high-glucose Dulbecco´s Modified Eagle´s Medium F12 (DMEM-F12) (Biosera, Kansas City, MO, United States). The growth medium was supplemented with a 10% foetal bovine serum, 1x HyClone™ Antibiotic/Antimycotic solution (GE Healthcare, Little Chalfont, UK), Epidermal growth factor (EGF) (20 ng/mL final), Hydrocortisone (0.5 μg/mL final) and Insulin (10 μg/mL final) (Sigma).

BJ-5ta (immortalised foreskin fibroblasts) were obtained from ATCC and cultured in Dulbecco´s Modified Eagle´s Medium (DMEM) supplemented with M199 medium (4:1), Hygromycin B (0.01 mg/mL) and 10% of foetal bovine serum.

Primary human umbilical cord vein endothelial cells (HUVECs) were isolated from umbilical cords obtained from the local hospital under P.J. Šafárik University in Košice. The study was approved by the Ethical Committee of the Faculty of Pharmacy, Comenius University, in Bratislava (06/2019). HUVECs were cultured in growth medium cM199 (= M199 medium supplemented with 20% heat-inactivated new-born calf serum, 10% heat-inactivated human serum, 150 μg/mL crude endothelial cell growth factor (ECGF), 5 U/mL heparin, 100 U/mL penicillin and 100 μg/mL streptomycin). Cells were cultured in an atmosphere containing 5% CO_2_ in humidified air at 37 °C.

### 2.5. Experimental Design

The MCF-10A and BJ-5ta cells were seeded and cultivated for 24 h in a complete medium. One day after seeding, 30 ng/mL of TGF-β was added and cultivated for 7 days (d). TGF-β was added to induce epithelial–mesenchymal transition in tested cell lines. Lichen extract *P. furfuracea* (L.) Zopf and its metabolite physodic acid were added at time 0 h and then incubated for 24, 48 and 72 h for various analyses, as described in Figure 1.

### 2.6. MTS (Methyl Tetrazolium Salt) Cell Viability Assay

Growth inhibitory activities of PSE and Phy toward MCF-10A (5 × 10^3^/well), fibroblasts (BJ-5ta) (5 × 10^3^/well) and HUVECs (6 × 10^3^/well) were determined by the MTS (3-(4,5-dimethylthiazol-2-yl)-5-(3-carboxymethoxyphenyl)-2-(4-sulfophenyl)-2H-tetrazolium) colorimetric assay. Briefly, MCF-10A and BJ-5ta were seeded in 96-well culture plates, and after 24 h incubation, treated with various ranges of concentrations of tested compounds (10–100 μg/mL) for 72 h. At the same time, HUVECs were seeded in a gelatine-coated 96-well plate and treated with the same aliquots of indicated compounds in the presence or absence of VEGF (Vascular endothelial growth factor) (25 ng/mL) for 48 h. Cell viability was evaluated by measuring the absorbance at 490 nm using the automated CytationTM 3 Cell Imaging Multi-Mode Reader (Biotek, Winooski, VT, USA). Three independent experiments were performed at different intervals. The inhibitory concentrations IC_10_ and IC_50_ values were calculated from these data.

### 2.7. 5-Bromo-2′-deoxyuridine (BrdU) Cell Proliferation Assay

The MCF-10A (5 × 10^3^/well), BJ-5ta (5 × 10^3^/well) and HUVEC cells (4 × 10^3^/well) were plated in a 96-well plate in 80 μL suitable medium. HUVECs were seeded onto a gelatine-coated plate. After 24 h, cells were treated with the range of concentration 10–100 μg/mL for 72 h (MCF-10A, BJ-5ta), or in the presence or absence of 25 ng/mL of VEGF for 48 h (HUVECs). After 48 h (MCF-10A, BJ-5ta) or 24 h (HUVECs), BrdU labelling solution was added into cells and incubated for another 24 h at 37 °C, followed by fixation and incubation with anti-BrdU peroxidase conjugate solution for an additional 1.5 h at room temperature (RT). Then, cells were washed with washing buffer PBS and incubated with substrate solution TMB for 5 to 30 min according to colour intensity. Finally, we added stop solution (1 M H_2_SO_4_) and incorporated BrdU was detected with an automated CytationTM 3 Cell Imaging Multi-Mode Reader at 450 nm (reference wavelength: 690 nm). Three independent experiments were performed at different intervals. The IC_10_ and IC_50_ values were calculated from these data.

### 2.8. Flow Cytometry Protein Analyses and Cell Cycle

The MCF-10A cells (1 × 10^4^/well) were seeded in Petri dishes and stimulated with 30 ng/L TGF-β for 7 days. After stimulation, transformed cells were treated with IC_10_ concentrations of tested compounds (PSE, Phy) for other 24, 48 and 72 h. After detaching, using Trypsin/EDTA, cells were harvested and pelleted by centrifugation at 1200 rpm for 5 min. Pellet was re-suspended in PBS and divided for a particular analysis. Cells were stained prior to analyses with E-Cadherin Mouse mAb (Alexa Fluor 488 conjugate) or N-cadherin Rabbit mAb (Alexa Fluor 647 conjugate) (Cell Signalling) for 15 min at RT in the dark. For the cell cycle analyses, cells were re-suspended in staining solution (final concentration 0.1% Triton X-100, 0.5 mg/mL ribonuclease A and 0.025 mg/mL propidium iodide (PI)), and incubated in the dark at RT for 30 min. All samples were analysed using a FACS Calibur flow cytometer (Becton Dickinson, San Jose, CA, USA). The IgG isotype control checking was performed for every used antibody.

### 2.9. Western Blot

Proteins isolated from MCF-10A and BJ-5ta cell lysates were determined by the Pierce^®^ BCA Protein Assay Kit (Thermo Scientific, Rockford, IL, United States), using bovine serum albumin (BSA) as the standard, and measured by an automated Cytation™ 3 Cell Imaging Multi-Mode Reader (Biotek) at a wavelength of 570 nm. Proteins were separated on SDS-PAA gel (12%) at 100 V for 2 h and then transferred to a polyvinylidene difluoride (PVDF) membrane using the iBlot dry blotting system (Thermo Scientific, Rockford, IL, United States). The membrane with the transferred proteins was blocked in 5% BSA in TBS (Tris-buffered saline)-Tween (pH 7.4) for 1 h at RT to minimise non-specific binding. The transferred membrane was subsequently incubated at 4 °C overnight with primary antibodies (Table 1). The next day, the membrane was washed in TBS-Tween (3 × 5 min) and incubated with the horseradish peroxidase (HRP)-conjugated anti-rabbit or anti-mouse secondary antibody (1:1000 dilution) for 1 h at RT. After incubation, the membrane was again washed in TBS-Tween (3 × 5 min), and the expression of the protein was detected using a chemiluminescent ECL substrate (Thermo Fisher Scientific) and MF-ChemiBIS 2.0 Imaging System (DNR BIO-Imaging Systems, Jerusalem, Israel). The detected band was then analysed densitometrically using the Image Studio Lite software (LI-COR Biosciences, Lincoln, NE, USA). Equal loading was verified using the antibodie β-actin. Detection was performed in triplicate.

### 2.10. Immunofluorescence Microscopy

The MCF-10A cells at a density of 2 × 10^3^/cm^2^, plated on 22 mm^2^ glass coverslips, were stimulated for 7 days with 30 ng/mL of TGF-β. On day 8, cells were treated with IC_10_ concentration of PSE and Phy in the presence or absence of TGF-β for 72 h. After treatment, cells were washed with PBS, fixed with 4% paraformaldehyde (pH 7.2) for 10 min, permeabilised by 0.1% Triton X-100 (Sigma-Aldrich) for 10 min at RT and then washed three) times with PBS for 5 min. Glass coverslips were then incubated with Swine serum (1:30) for 30 min for blocking non-specific binding. Primary antibodies were diluted according to the manufacturer’s instructions (Table 2) and transferred to coverslips for 90 min. Coverslips were washed three times in PBS and incubated with secondary antibodies. Cells’ nuclei were stained by 4′,6-diamidino-2-phenylindole (DAPI, Sigma-Aldrich). All coverslips were mounted in Vectashield (Cole-Parmer, Illinois, USA) and immunofluorescence pictures were taken with the same exposition settings by the Nikon Eclipse 90i fluorescence microscope (Nikon, Tokyo, Japan) equipped with filter cubes for FITC and DAPI. Images were then analysed using ImageJ (NIH) software.

### 2.11. The Chorioallantoic Membrane (CAM) Assay

Fertilised quail eggs (*Coturnix coturnix japonica*; 20 specimens) were obtained from the certified farm (Mala Ida, Slovakia) and incubated horizontally in a forced draft constant humidity incubator at 38.2 ± 0.5 °C and 58% relative humidity. After 56 h, the eggs were sterilised by wiping the surface lightly with 70% ethanol and allowed to dry. An incision was made in the middle ventral part of the shell, using sterile scissors, and the embryos were deposited in six-well tissue culture dishes (Sigma-Aldrich), which were then returned to the humidified incubator for an additional 4 days, until the start of the experiments. At day 7 (96 h after day 3), a sterilised silicone ring (10 mm inner diameter) was laid on CAM surface for the deposition of testing solution. The control group was treated with Sodium Chloride 0.9% (30 μL per egg), while the positive control group was treated with IC_10_ values of PSE and Phy in the presence or absence of 25 ng/mL of VEGF. At least 10 eggs were used in every experimental group. The photographs of CAM blood vessels’ formation inside of rings were obtained using a stereomicroscope Olympus SZ61 (Tokyo, Japan) and digital camera PROMICRA 3.2 (Prague, Czech Republic). Subsequently, photographs were pre-processed using QuickPHOTO MICRO microscope software (Promicra; Prague, Czech Republic). Representative images were then analysed using Wimasis Image Analysis automatic software to quantify the angiogenesis.

### 2.12. Statistical Analyses

Results are expressed as mean ± standard deviation (SD). Statistical analyses of the data were performed using standard procedures, with one-way analysis of variance (ANOVA) followed by the Bonferroni multiple comparisons test. Differences were considered significant when *p* < 0.05. Throughout this paper * indicates *p* < 0.05, ** *p* < 0.01, *** *p* < 0.001 versus untreated control; Δ *p* < 0.05, ΔΔ *p* < 0.01, ΔΔΔ *p* < 0.001 compared to TGF-β; # *p* < 0.05, ## *p* < 0.01, ### *p* < 0.001 versus VEGF.

## 3. Results

### 3.1. HPLC

Five fractions of lichen secondary metabolites were isolated in the sample of lichen *Pseudevernia furfuracea* (L.) Zopf. Based on the internal standards, atranorin, chloratranorin, physodalic acid, 3-hydroxyphysodic and physodic acid were identified by HPLC (Figure 2). In the following experiments, the whole *Pseudevernia furfuracea* (L.) Zopf extract and third potent metabolite physodic acid were used.

### 3.2. NMR

The structure of physodic acid (Appendix A) was fully characterised by one-dimensional (1D) total correlation spectroscopy (TOCSY, Appendix A), ^1^H, ^1^H correlation spectroscopy (COSY, Appendix A), ^1^H, ^13^C heteronuclear single-quantum coherence (HSQC, Appendix A) and ^1^H, ^13^C heteronuclear multiple-bond correlation (HMBC, Appendix A) spectra. The aliphatic region of the 1H NMR spectra (Appendix A) contain typical multiplets corresponding to protons H-1″–H-7″ and H-1ʺ–H-5ʺ of alkyl chains. The remaining three proton signals appear in the typical aromatic proton frequency range (*δ*_H_ 6.55 (1H, d, *J* = 2.3 Hz, H-5), 6.63 (1H, s, H-3′) and *δ*_H_ 6.64 (1H, d, *J* = 2.4 Hz, H-3)) (Appendix A). HSQC connectivities (Appendix A) were used to identify protonated carbons and HMBC connectivities (Appendix A) were used to assign non-protonated carbons.

### 3.3. MTS Cytotoxic Assay

The cell viability of PSE and Phy was evaluated on three different types of cell line (MCF-10A (Figure 3A), BJ-5ta (Figure 3B) and HUVECs (Figure 3C, D) using the MTS assay, as described in the Material and Methods Section. Studied compounds were dissolved in dimethylsulfoxide (DMSO). The final concentration of DMSO in the culture medium was <0.2% and exhibited no cytotoxicity, as displayed in Figure 3A–C. As shown in Figure 3, various cell lines displayed different sensitivity to PSE and Phy, with IC_10_ values ranging from 22.10 ± 1.82 to 35.79 ± 1.40 μM for PSE and 46.35 ± 0.39 to 60.62 ± 3.30 μM for Phy treatment. The IC_50_ values ranged from 45.53 ± 0.11 to 92.70 ± 4.32 μM for PSE and 79.88 ± 0.93 to 92.76 ± 3.62 μM for Phy treatment (Table 3). The most resistant cell line on PSE and Phy treatment was identified as the HUVEC cell line, followed by BJ-5ta fibroblasts and breast MCF-10A cells. Moreover, results revealed that PSE and Phy significantly inhibited VEGF-induced cell growth, while no significant differences were observed in non-stimulated HUVECs (Figure 3C,D). To detect the potential anti-proliferative effect of both compounds, the BrdU proliferation assay was used.

### 3.4. Bromdeoxyuridine (BrdU) Incorporation

The antiproliferative effect of tested compounds was measured with the thymidine analogue BrdU (5-bromo-2-deoxyuridine) following its incorporation into newly synthetised DNA. As mentioned above, studied substances were dissolved in DMSO and exhibited no cytotoxicity, as shown in Figure 4A–C. Our results (Figure 4) showed that BrdU demonstrated higher values of IC_10_ (37.49 ± 0.82 to 59.84 ± 4.50 μM for PSE, and 62.88 ± 0.90 to 89.53 ± 3.21 μM for Phy treatment) and IC_50_ (83.80 ± 3.48 to 116.82 ± 1.82 μM for PSE, and 124.16 ± 5.33 to 164.26 ± 4.50 μM for Phy treatment) (Table 4) compared to the MTS cytotoxic assay. There are no significant differences in non-stimulated HUVECs compared to VEGF-stimulated HUVECs, which correlates with results obtained from MTS. For further analyses, IC_10_ and IC_50_ values of BrdU were used.

### 3.5. Effect of IC_10_ Concentration of PSE and Phy on MCF-10A Cell Proliferation

In order to investigate the effect of PSE and Phy on the cell cycle phase distribution in MCF-10A cells, flow cytometry analysis was assessed. MCF-10A were treated with IC_10_ concentration of PSE and Phy in the absence or presence of TGF-β (30 ng/mL) for 24, 48 and 72 h. As shown in Figure 5 and Table 5, there are no significant differences between control and PSE/Phy-treated cells as well as between TGF-β-stimulated cells and those treated with PSE/Phy, which indicates normal proliferative activity in low-dose treatments.

### 3.6. N-Cadherin Regulation Analyses after PSE and Phy Treatment

The aberrant expression of N-cadherin is a hallmark of epithelial–mesenchymal transition [41], which plays an important role in tumour invasion, metastasis and relapse [8]. Therefore, in this study, we investigated how PSE and Phy contributed to EMT. The epithelial breast MCF-10A cells were initially stimulated by TGF-β (30 ng/mL) for 7 days and then exposed to tested compounds, as illustrated in Figure 1A. As expected, stimulation by TGF-β led to increasing expression of mesenchymal marker N-cadherin and to morphological changes of MCF-10A cells, as shown in Figure 1B. To detect wheter PSE and Phy are capable to modulate EMT, we first evaluated the time dependence of the tested substances using flow cytometry analysis. As displayed in Figure 6C,D, both PSE and Phy significantly reduced N-cadherin in TGF-β-stimulated MCF-10A cells as time progresed, compared to TGF-β alone. Since their effects reached significant changes at 48 and 72 h, for further investigation, 24 h was not used. To detect a possible dose dependence effect of PSE/Phy treatment, we performed Western blot analysis by using IC_10_ and IC_50_ of both substances. As shown in Figure 6A,B, there is a significant change in N-cadherin expression in a dose-dependent manner. Since there is also a significant effect of PSE using IC_10_ concentration, we decided to perform immunofluorescence staining of N-cadherin (Figure 6E) with lower concentrations of the tested compounds, while we wanted to observe maximal cell viability of MCF-10A cells. Treatment with IC_10_ of PSE and Phy led to a significant decrease of N-cadherin expression at 72 h in TGF-β-stimulated MCF-10A cells compared to TGF-β alone, which points to the possibility of PSE and Phy to regulate EMT in low concentrations.

### 3.7. E-cadherin Regulation Analyses after PSE and Phy Treatment

Epithelial cells are characterised by having specialised cell-to-cell junctions and adhesion proteins, such as E-cadherin. During EMT, components of cell–cell junctions become inactive, which express in downregulation of E-cadherin [42]. In this work, we demonstrated that non-transformed epithelial breast MCF-10A cells were associated with high levels of E-cadherin expression, which we proved using Western blot analysis (Figure 7A,B), flow cytometry analysis (Figure 7C,D) and by immunofluorescence staining (Figure 7E). Stimulation with TGF-β, as the main inducer of EMT [16], led to decreased expression of E-cadherin, as shown in the aforementioned figures. However, we did not observe a significant restoration of E-cadherin after PSE or Phy treatment in TGF-β-stimulated groups. The E-cadherin level did not change at all compared to control (untreated) and TGF-β alone.

### 3.8. Fibronectin Expression Analyses after PSE and Phy Treatment

Numerous studies have demonstrated that aberant expression of fibronectin is associated with poor prognosis in patients with invasive breast cancer [43,44]. Therefore, we performed Western blot and immunofluorescence analyses of fibronectin in TGF-β-stimulated MCF-10A cells as well as to confirm the inhibition effect of PSE and Phy on mesenchymal markes. Immunofluorescence (IF) imaging of fibronectin (Figure 8A) revealed that its expression is associated with TGF-β, while treatment with tested substances led to fibronectin downregulation. Western blot analysis (Figure 8B) confirmed data obtained from IF staining. Moreover, PSE and Phy downregulated levels of fibronectin in a dose-dependent manner.

### 3.9. EMT-Associated Proteins’ Expression Analyses after PSE and Phy Treatment in TGF-β-Stimulated BJ-5ta Fibroblasts

Fibroblasts within the tumour stroma of breast cancer, termed as cancer-associated fibroblasts, are a dominant component of TME and are found to induce EMT in cancer cells, which make them a great target for cancer therapy [45,46,47]. Fibroblasts can be reversibly or irreversibly activated in response to numerous cytokines, including TGF-β, secreted by cancer cells and other stroma cells [48]. To investigate the potential inhibitory role of PSE and Phy on CAFs in the present study, healthy immortalised foreskin fibroblasts (BJ-5ta) were transformed into CAFs by TGF-β for 7 days. Changes in the expression of E-cadherin, N-cadherin, fibronectin, alpha smooth muscle actin (α-SMA) and transcription factor Slug (Figure 9A–F) were examined by Western blot after 48 and 72 h of IC_10_ PSE and Phy treatment. Results obtained from BJ-5ta cells showed that PSE and Phy treatment significantly decrease mesenchymal markers in a time- and dose-dependent manner, but expression of epithelial marker E-cadherin did not significantly change, which correlates with MCF-10A cells.

Next, we studied the TGF-β-Smad signalling pathway, which is one of the most important pathways that induces CAF activation [49]. As displayed in Figure 9A,G, the phosphorylation levels of Smad2 and Smad3 (pSmad2/3) decreased after PSE and Phy treatment in a time-dose manner. Total Smad2/3 levels were altered minimally after PSE and Phy treatment.

### 3.10. Angiogenesis Analyses after PSE and Phy Treatment in a Quail Embryo CAM Model

In the present study, we evaluated an ex ovo model based on the chorioallantoic membrane (CAM) of fertilised quail eggs for explaining the mechanisms of PSE and Phy as potent inhibitors of tumour angiogenesis. Inhibition of angiogenesis was analysed using the WimCam software program, that gave us information about vessels’ density, total vessel network length and total branching points. The CAM images (Figure 10A) revealed the ability of VEGF to form pathological angiogenesis conditions, as expected. The present results (Figure 10B–D) demonstrate that both PSE and Phy showed a significant decrease in vascular density (Figure 10B), total vascular network length (Figure 10C) and total branching points (Figure 10D) in VEGF-stimulated HUVECs compared to VEGF alone after 72 h treatment. In a non-stimulated group, there is a significant change in vessel density after Phy treatment and in total vessels’ network length after PSE treatment. Total branching points were not significantly affected after PSE and Phy treatment in the absence of VEGF. None of the tested groups displayed any damage to blood vessels or embryo viability. Mortality of the embryos was minimum (10%) and did not depend on the drug dose treatment. During experimentation, we did not observe symptoms of haemorrhage or coagulation, which points to no toxicity of the tested compounds.

## 4. Discussion

Throughout history, natural plant compounds have played a dominant role in cancer treatment. Several of them, such as lichens, have been reported to possess a cytotoxic effect against various cancer cell lines, including breast cancer [50,51,52]. Lichens are naturally occurring symbiotic organisms that produce different secondary metabolites. Although there is no clinical study, many of them revealed a wide range of biological activities in vitro and in vivo, such as antioxidant, antiviral, antibiotics, antifungal and anticancer [37,53,54,55,56]. Numerous previous studies have observed anticancer activity of various secondary metabolites in different cancer cell lines, which indicate their possible use in cancer therapy [37,50,51,57,58,59]. However, only a few studies demonstrated the potential anticancer activity of extract *Pseudevernia furfuracea* (L.) Zopf and metabolite physodic acid against different cell lines [56,60,61,62,63]. Therefore, we decided to study mechanisms of antiproliferative action of these substances in more detail on three different cell lines (breast cells: MCF-10A, fibroblasts: BJ-5ta and endothelial cells: HUVECs).

The breast cancer (BC) incidence worldwide in women still remains high [1]. Triple-negative breast cancer represents 15–20% of all breast cancers, with the worst outcome among all BC subtypes. Moreover, this subtype is known to be a heterogeneous disease at the biological, clinical and genomic levels [64,65,66]. It is the tumour microenvironment that contributes significantly to breast cancer progression by re-programming the surrounding stromal cells (fibroblasts, endothelial cells, macrophages, etc.) and influences breast cancer behaviour and metastasis [67]. Therefore, the selection of experimental objects in this study was not random and we focused on the non-cancerous part of breast tumour stroma that could be a target of action for potential lichen metabolites with anticancer properties.

The results obtained from cell viability and cell proliferation tests indicated that both presented compounds (PSE and Phy) did not show a high degree of cytotoxicity on all tested cells and confirmed a differential sensitivity of studied cell lines to lichen secondary metabolites.

Several studies have demonstrated that EMT plays a key role in tumour development, which indicates that pharmacological interventions targeting this process may provide new therapeutic strategies for cancer therapy [8]. Among many other inducers, TGF-β is one of the strongest inducers of EMT [68] in several cell lines, including the human mammary epithelial cell line MCF-10A [69]. After activation of EMT, mesenchymal cells displayed a spindle-like elongated morphology [15], as shown in Figure 1B. Here, we demonstrated how seven-day exposure to TGF-β (30 ng/mL) allows epithelial cells (MCF-10A) to separate from one another and thus acquire motility phenotype. Furthermore, the EMT process is associated with an upregulation of mesenchymal marker N-cadherin followed by the downregulation of epithelial marker E-cadherin and is regulated by various signalling pathways and transcription factors [70].

Recently, Yang et al. [71] have proved that lichen extract *Pseudocyphellaria coriacea* and its metabolite physciosporin were capable to decrease the level of several EMT-associated proteins, such as N-cadherin, Snail and Twist, while changes in E-cadherin levels were not observed. Another study has revealed the inhibitory effect of extract from lichen *Flavocetraria cucullata* and metabolite usnic acid (UA) on the EMT process through decreasing the mRNA level of E-cadherin [72].

The present study has shown, for the first time, the effect of lichen extract *Pseudevernia furfruacea* (L.) Zopf and its metabolite physodic acid on the modulation of the tumour microenvironment. In this work, we demonstrated that PSE and Phy led to the initiation of EMT in TGF-β-stimulated MCF-10A cells by decreasing the expression of mesenchymal markers, N-cadherin and fibronectin, in a time- and dose-dependent manner. However, the epithelial properties were not markedly affected. Of note is that, in some case, changes in E-cadherin level are not essential for EMT or metastatic activity of tumour [73,74]. As mentioned above, the study observed that the effect of lichen extract *P. coriacea* on the EMT process [71] also did not display changes in E-cadherin expression, which is in accordance with our results. It is known in breast cancer that stromal-derived mesenchymal stem cells or mesenchymal-like cells are precursors of cancer-associated fibroblasts [75]. Therefore, inhibition or downregulation of mesenchymal markers’ expression after PSE and Phy treatment could be beneficial in the inhibition of breast cancer induction, progression or metastasis.

Among the components of TME of breast and other tumours, cancer-associated fibroblasts are one of the most crucial types of stromal cells which promote the growth of cancer cells by various mechanisms [76,77,78,79,80,81]. CAFs found in the TME are large spindle-shaped mesenchymal cells [82], characterised by expression of alpha-smooth muscle actin [83]. In the recent study, healthy immortalised foreskin fibroblasts (BJ-5ta) were transferred into CAFs by TGF-β, which has been verified using Western blot, where we found increasing levels of mesenchymal markers, including α-SMA. A subsequent treatment with experimental compounds resulted in a significant reduction of mesenchymal markers (N-cadherin, fibronectin, α-SMA) in a time-dependent manner. The protein levels of E-cadherin after PSE and Phy treatment correlated with those obtained from MCF-10A cells after 72 h treatment.

Next, we studied the TGF-β canonical pathway Smad2/3. The altered protein levels of Smad2 and Smad3 were observed in several type of cancers, including breast cancer, compared to normal mammary epithelial cells [84]. A recent study demonstrated that a high level of pSmad2 in breast cancer cells was associated with poor breast cancer prognosis [85]. Another study revealed that blocking the function of Smad3 in the MCF-10A-derived breast cancer cells led to suppression of metastatic foci in lungs of mice [86,87]. The same study demonstrated that overexpression of Smad3 enhanced malignancy of primary tumours [87]. Therefore, downregulation of these proteins could be a potential target for therapeutics. In accordance with the above-mentioned data, the analysis of downstream phospho Smad2 and Smad3 revealed decreased phosphorylation after PSE and Phy treatment compared to TGF-β. In tumour cells, the TGF-β pathway also induces Slug, Snail, Twist and ZEB expression via the Smad proteins [88]. Among these factors, Slug expression was evaluated using Western blot analysis in the present study. The 72 h PSE and Phy treatment resulted in downregulation of Slug protein, which correlates with Smad2/3 expression.

Additionally, the current study revealed the anti-angiogenic efficiency of PSE and Phy for the first time. To the best of our knowledge, there is no information about the anti-angiogenic activity of PSE and Phy in vitro or in vivo. It has been reported that some lichen secondary metabolites such as usnic acid, vulpinic acid and olivetoric acid possess anti-angiogenic activity. Koparal [89] reported that usnic acid and vulpinic acid exhibited strong anti-angiogenic activity by inhibiting endothelial tube formation. Moreover, Song et al. [90] reported that (+)- usnic acid inhibits VEGF-induced HUVEC migration and tube formation as well as VEGF-mediated signalling pathways such as AKT and ERK. The study also demonstrated that UA inhibits angiogenesis in vivo in the chorioallantoic membrane assay and in the corneal angiogenesis assay. The anti-angiogenic activity of olivetoric acid was also demonstrated by Koparal et al. [91] by inhibiting endothelial tube formation.

Since angiogenesis is involved in the progression and metastasis of various human cancers [92], it represents another important therapeutic target in our research. Therefore, we conducted an ex ovo CAM assay to examine the anti-angiogenic effect of PSE and Phy on vascularisation induced by VEGF. The CAM model, as an experimental platform for the study of blood vessels, has become highly desirable for research purposes over the last two decades [93]. The greatest advantage of this model is that there is no conflict with ethical and legal obligations compared to other animal models, as the experiments begin and end before hatching [94]. This study clearly demonstrated an anti-angiogenic role of PSE and Phy that was indicated by vessel density, total vessel network length and total branching points. The CAM assay showed that PSE and Phy significantly reduced all tested parameters at IC_10_ concentrations.

## 5. Conclusions

Altogether, findings of the current study, for the first time, demonstrated that lichen extract *Pseudoevernia furfuracea* (L.) Zopf and one of its metabolites, physodic acid, can modulate the tumour microenvironment. The results showed that lichen extract regulated TME more strongly than physodic acid alone since there could have been a synergistic effect of the individual substances obtained in PSE. Moreover, tested compounds were able to modulate TME in very low concentrations (IC_10_), which allows preserving the cell viability of tested cell lines. Both PSE and Phy treatments alone or in TGF-β-stimulated MCF-10A or BJ-5ta cells were able to modulate several EMT-associated proteins, such as N-cadherin, fibronectin, α-SMA, Slug and Smad2/3, in a concentration- and time-dependent manner. Moreover, PSE and Phy altered the angiogenesis process, as confirmed by the CAM assay. These findings uncover the novel mechanisms of PSE and Phy in TME as a potential cancer therapy target. However, future studies might be needed to determine more associated mechanisms of the anticancer action of the tested substances more deeply. Despite the perspective of lichen extracts or isolated lichen molecules as anticancer or tumour environment modulating substances, currently, no relevant clinical research provides relevant data in humans. Moreover, some limitations such as rarity in nature, extraction and purity issues exist. However, the chemical structure of most lichen molecules is simple, which promotes their easy synthesis. In this regard, many of these synthetic substances may be applied and provide reasonable clinical use.

## Figures and Tables

**Figure 1 biomolecules-11-00420-f001:**
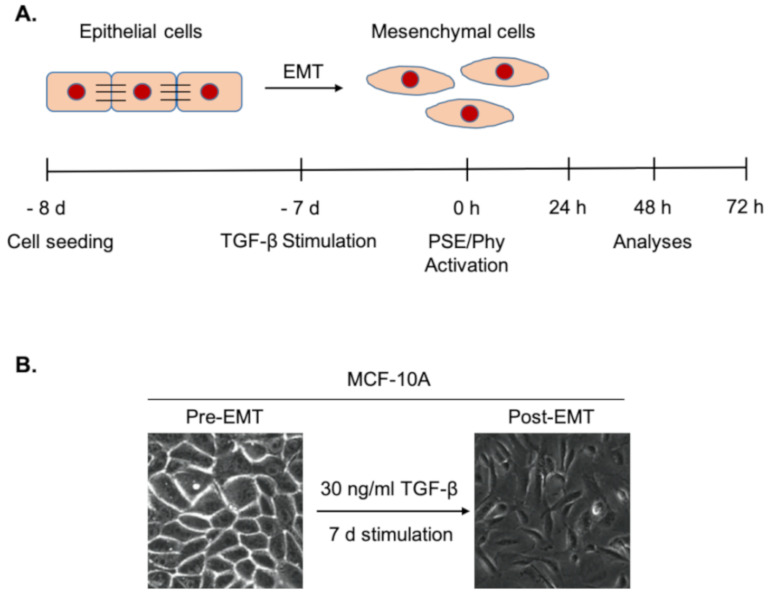
(**A**) Experimental design for study epithelial to mesenchymal transition (EMT) in MCF-10A and BJ-5ta cells and (**B**) EMT morphological changes after 7 days (d) of TGF-β stimulation. Original magnification, 40x. Copyright Petrova 2020.

**Figure 2 biomolecules-11-00420-f002:**
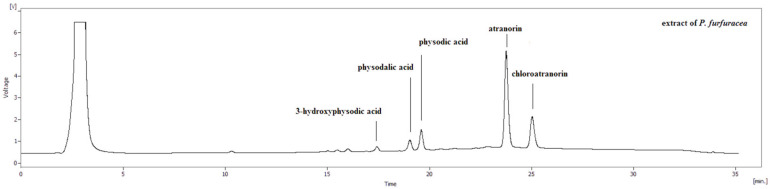
High-performance liquid chromatography (HPLC) chromatograph of identified lichen compounds from extract *P. furfuracea* (L.) Zopf.

**Figure 3 biomolecules-11-00420-f003:**
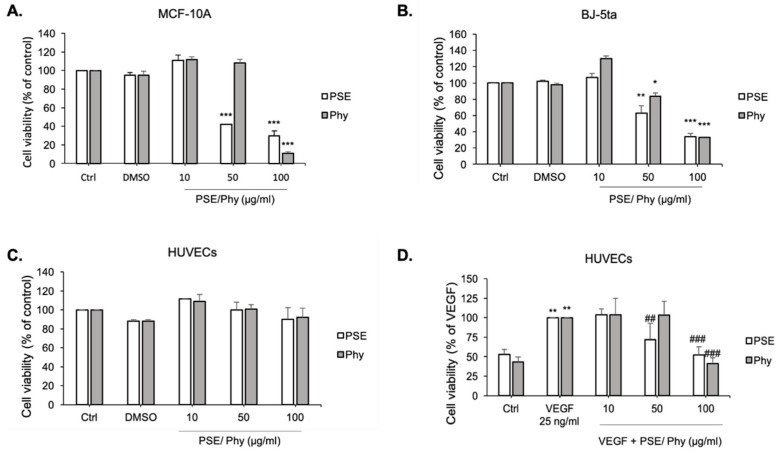
Effect of *Pseudevernia furfuracea extract* (PSE) and Physodic acid (Phy) on MCF-10A, BJ-5ta and HUVEC cells’ viability. MCF-10A (**A**) and BJ-5ta (**B**) cells were treated with various concentrations of PSE and Phy (10–100 μg/mol) for 72 h. HUVECs were treated with the same range of concentrations of tested compounds in the absence (**C**) or presence (**D**) of VEGF (Vascular endothelial growth factor) (25 ng/mL) for 48 h. Values represent the mean ± standard deviation. (* *p* < 0.05, ** *p* < 0.01, *** *p* < 0.001 compared to control; ## *p* < 0.01, ### *p* < 0.001 compared to VEGF based on Student´s t-test).

**Figure 4 biomolecules-11-00420-f004:**
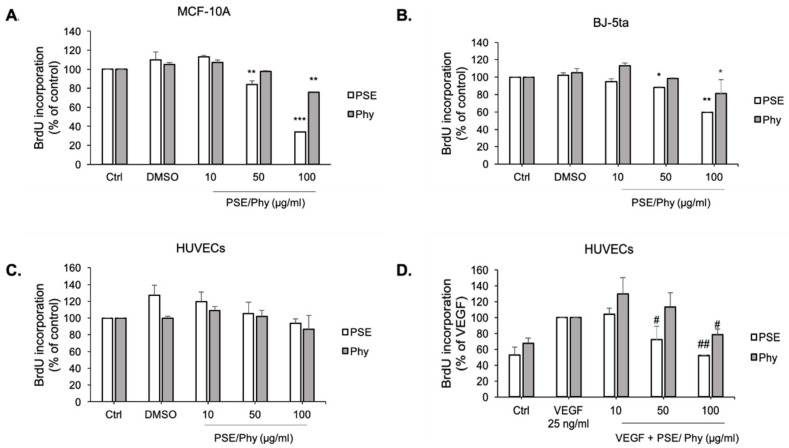
Effect of PSE and Phy on MCF-10A, BJ-5ta and HUVECs BrdU (5-bromo-2'-deoxyuridine) incorporation. MCF-10A (**A**) and BJ-5ta (**B**) cells were treated with various concentrations of PSE and Phy (10–100 μg/mol) for 72 h. HUVECs were treated with the same range of concentrations of tested compounds in the absence (**C**) or presence (**D**) of VEGF (25 ng/mL) for 48 h. Values represent the mean ± standard deviation. (* *p* < 0.05, ** *p* < 0.01, *** *p* < 0.001 compared to control; # *p* < 0.05, ## *p* < 0.01 compared to VEGF).

**Figure 5 biomolecules-11-00420-f005:**
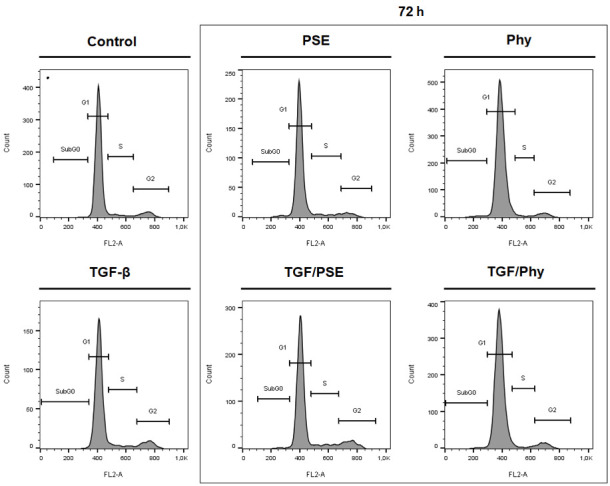
Representative diagrams for cell cycle phase distribution in MCF-10A cells. Cells were exposed to 30 ng/mL of TGF-β (Transforming growth factor beta) for 7 days and subsequently stimulated with PSE and Phy for 72 h.

**Figure 6 biomolecules-11-00420-f006:**
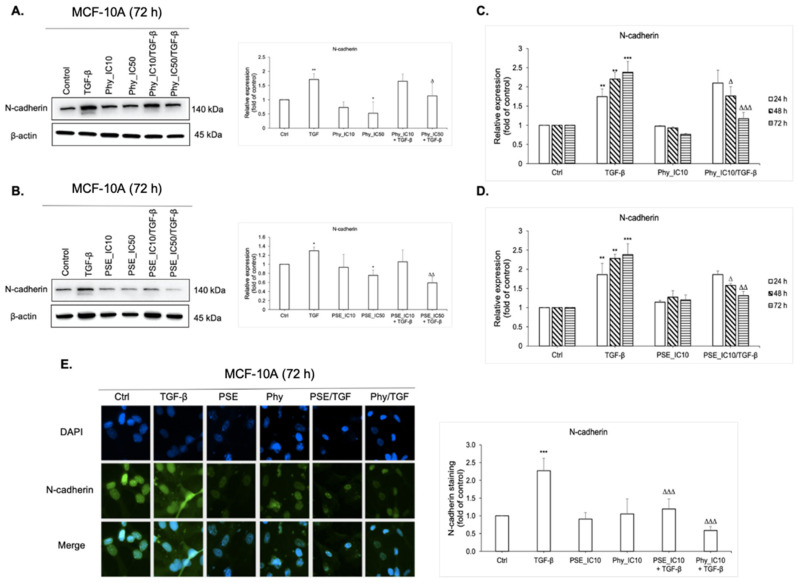
PSE and Phy inhibition effect on epithelial to mesenchymal transition (EMT) through decreasing N-cadherin expression in MCF-10A cells. The protein levels of EMT mesenchymal marker N-cadherin were assayed by Western blot analysis using IC_10_ and IC_50_ concentrations of Phy (**A**) and PSE (**B**). The band intensities of target protein were analysed using the Image Studio Lite software. Quantification is relative to the control and normalised to β-actin expression. Flow cytometry analysis of N-cadherin expression after 24, 48 and 72 h treatment with IC_10_ Phy (**C**) and PSE (**D**). (**E**) Representative fluorescence microscopy images of N-cadherin expression in MCF-10A cells after IC_10_ PSE and Phy treatment. The green signal represents corresponding protein staining by Alexa Fluor 488 and the blue signal indicates nuclei staining by DAPI. Original magnification x200. The fluorescence staining of N-cadherin was analysed using ImageJ software. Quantification is relative to the control. Data in graphs are shown as mean ± standard deviation (SD) for three separate experiments. * *p* < 0.05, ** *p* < 0.01, *** *p* < 0.001 compared to control; Δ *p* < 0.05, ΔΔ *p* < 0.01, ΔΔΔ *p* < 0.001 compared to TGF-β.

**Figure 7 biomolecules-11-00420-f007:**
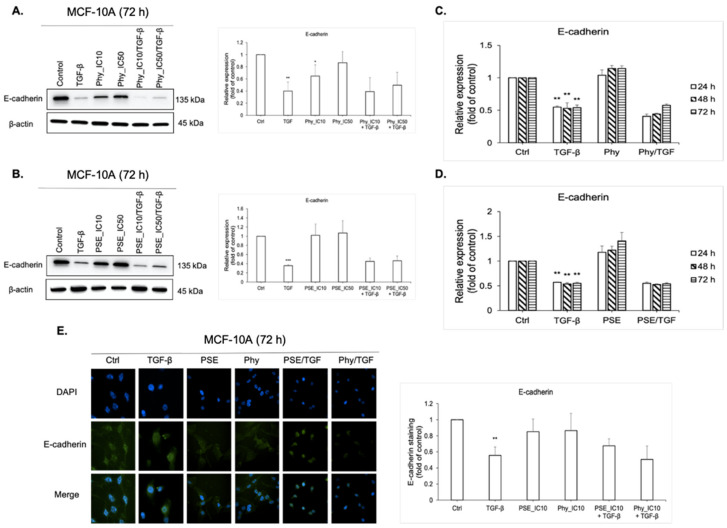
E-cadherin expression in MCF-10A cells after PSE and Phy treatment. The protein levels of EMT epithelial marker E-cadherin were assayed by Western blot analysis using IC_10_ and IC_50_ concentrations of Phy (**A**) and PSE (**B**). The band intensities of target protein were measured using the Image Studio Lite software. Quantification is relative to the control and normalised to β-actin expression. Flow cytometry analysis of E-cadherin expression after 24, 48 and 72 h treatment of IC_10_ of Phy (**C**) and PSE (**D**). (**E**) Representative fluorescence microscopy images of E-cadherin expression in MCF-10A cells after 72 h treatment of IC_10_ PSE and Phy. The green signal represents corresponding protein staining by Alexa Fluor 488 and the blue signal indicates nuclei staining by DAPI. Original magnification x200. The fluorescence staining of N-cadherin was analysed using ImageJ software. Quantification is relative to the control. Data in graphs are shown as mean ± SD for three separate experiments. * *p* < 0.05, ** *p* < 0.01, *** *p* < 0.001 compared to control; Δ *p* < 0.05, ΔΔ *p* < 0.01, ΔΔΔ *p* < 0.001 compared to TGF-β.

**Figure 8 biomolecules-11-00420-f008:**
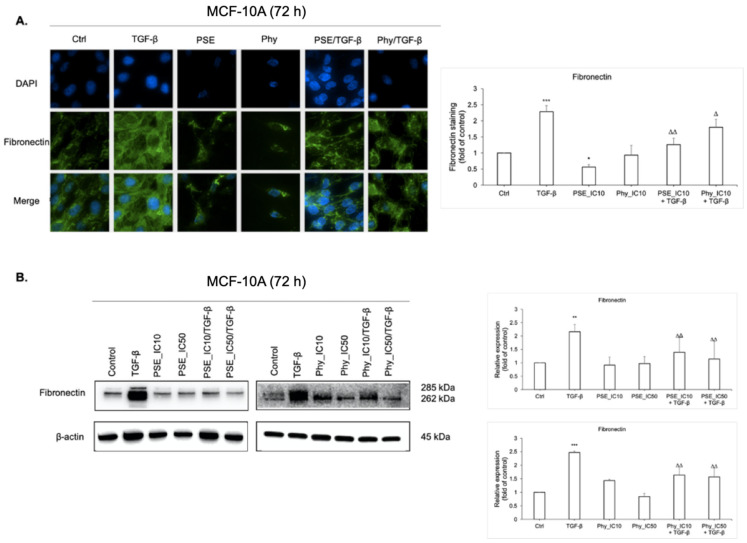
PSE and Phy reduced expression of fibronectin. (**A**) Immunofluorescence detection of fibronectin in MCF-10A cells. The green signal represents corresponding protein staining by Alexa Fluor 488 and the blue signal indicates nuclei staining by DAPI. Original magnification x200. The fluorescence staining of fibronectin was analysed using ImageJ software. Quantification is relative to the control. The protein levels of mesenchymal marker fibronectin were assayed by Western blot analysis (**B**) using IC_10_ and IC_50_ concentrations of PSE and Phy. The band intensities of target protein were measured using the Image Studio Lite software. Quantification is relative to the control and normalised to β-actin expression. Data in graphs are shown as mean ± SD for three separate experiments. * *p* < 0.05, ** *p* < 0.01, *** *p* < 0.001 compared to control; Δ *p* < 0.05, ΔΔ *p* < 0.01 compared to TGF-β.

**Figure 9 biomolecules-11-00420-f009:**
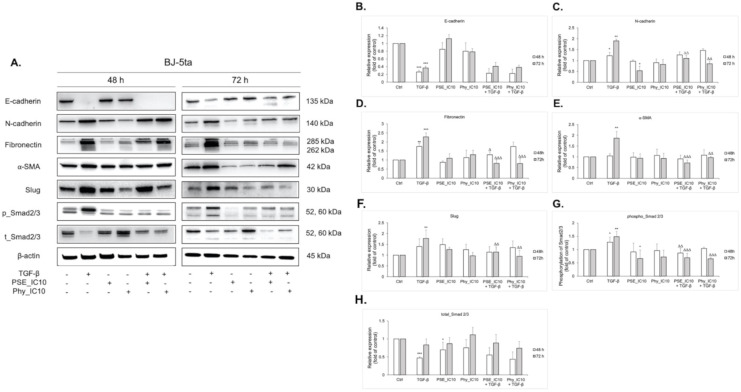
Analysis of the expression of EMT markers in BJ-5ta cells. (**A**) BJ-5ta cell lysates from treated or untreated cells were immunoblotted with different antibodies. The band intensity of E-cadherin (**B**), N-cadherin (**C**), fibronectin (**D**), alpha smooth muscle actin (α-SMA) (**E**), Slug (**F**), p-Smad2/3 (**G**) and t_Smad2/3 (**H**) was quantified using the Image Studio Lite software. Quantification is relative to the control and normalised to β-actin expression (* *p* < 0.05, ** *p* < 0.01, *** *p* < 0.001 compared to control; Δ *p* < 0.05, ΔΔ *p* < 0.01, ΔΔΔ *p* < 0.001 compared to TGF-β).

**Figure 10 biomolecules-11-00420-f010:**
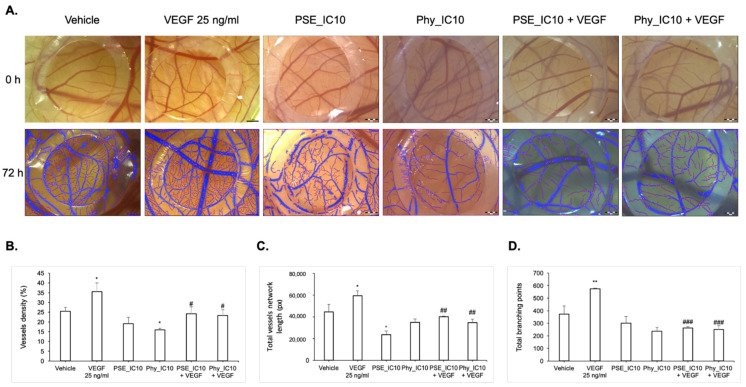
Effects of PSE and Phy on blood vessel formation in the *ex ovo* CAM (chorioallantoic membrane) assay. (**A**) Representative digital images of the most effective 72 h treatment with PSE and Phy in the presence or absence of VEGF (25 ng/mL). Quantitative analysis of angiogenesis by measuring vessels’ density (**B**), total vessels’ network length (**C**) and total branching points (**D**) based on the WimCAM data. Error bars represent mean ± SD (* *p* < 0.05, ** *p* < 0.01 versus vehicle; # *p* < 0.05, ## *p* < 0.01, ### *p* < 0.001 versus VEGF alone).

**Table 1 biomolecules-11-00420-t001:** List of Western blot antibodies.

Primary Antibodies	Mr (kDa)	Origin	Dilution	Source
Anti-Fibronectin antibody (ab2413)	285, 262	Rabbit	1:1000	Abcam
N-Cadherin (D4R1H) XP Rabbit mAb	140	Rabbit	1:1000	Cell Signaling Technology, Danvers, Massachusetts, United States
E-Cadherin (4A2) Mouse mAb	135	Mouse	1:1000	
α-Smooth Muscle Actin (D4K9N) XP Rabbit mAb	42	Rabbit	1:1000	
Slug (C19G7) Rabbit mAb	30	Rabbit	1:1000	
Phospho_Smad2 (Ser465/467)/Smad3 (Ser423/425) (D27F4)	52, 60	Rabbit	1:1000	
Smad2/3 Antibody	52, 60	Rabbit	1:1000	
β-Actin (8H10D10) Mouse mAb	45	Mouse	1:2500	
**Secondary antibodies**				
Anti-Rabbit IgG HRP	-	Goat	1:1000	Cell Signaling Technology, Danvers, Massachusetts, United States
Anti-Mouse IgG HRP	-	Goat	1:1000	

**Table 2 biomolecules-11-00420-t002:** List of antibodies for immunofluorescence staining.

Primary Antibodies	Origin	Dilution	Source
Anti-Fibronectin antibody (ab2413)	Rabbit	1 μg/mL	Abcam
N-Cadherin (D4R1H) XP Rabbit mAb	Rabbit	1:200	Cell Signaling Technology, Danvers, Massachusetts, United States
E-Cadherin (4A2) Mouse mAb	Mouse	1:200	
**Secondary antibodies**			
Alexa Fluor 488 anti-mouse IgG	Goat	1:1000	Cell Signaling Technology, Danvers, Massachusetts, United States
Alexa Fluor 488 anti-rabbit IgG	Goat	1:1000	

**Table 3 biomolecules-11-00420-t003:** The inhibitory concentrations IC_10_ and IC_50_ values of *Pseudevernia furfuracea extract (*PSE) and Physodic acid (Phy) for MCF-10A, BJ-5ta and HUVEC cells.

Cell Line	*Pseudevernia furfuracea* (L.) Zopf Extract		Physodic Acid	
	IC_10_ (μM)	IC_50_ (μM)	IC_10_ (μM)	IC_50_ (μM)
**MCF-10A**	22.10 ± 1.82	45.53 ± 0.11	59.25 ± 1.36	79.88 ± 0.93
**BJ-5ta**	26.60 ± 1.81	72.22 ± 9.11	46.35 ± 0.39	83.25 ± 1.02
**HUVECs**	35.79 ± 1.40	92.70 ± 4.32	60.62 ± 3.30	92.76 ± 3.62

**Table 4 biomolecules-11-00420-t004:** IC_10_ and IC_50_ values of PSE and Phy for MCF-10A, BJ-5ta and HUVEC cells.

Cell Line	*Pseudevernia furfuracea* (L.) Zopf Extract		Physodic Acid	
	IC_10_ (μM)	IC_50_ (μM)	IC_10_ (μM)	IC_50_ (μM)
**MCF-10A**	38.72 ± 2.25	83.80 ± 3.48	63.63 ± 3.40	155.00 ± 7.40
**BJ-5ta**	37.49 ± 0.82	116.82 ± 1.82	62.88 ± 0.90	164.26 ± 4.50
**HUVECs**	59.84 ± 4.50	110.52 ± 2.10	89.53 ± 3.21	124.16 ± 5.33

**Table 5 biomolecules-11-00420-t005:** Cell cycle phase distribution of MCF-10A cells in the presence or absence of 30 ng/mL TGF-β and PSE or Phy for 24, 48 and 72 h.

Treatment	Time (h)	Control	TGF-β	PSE	Phy	PSE/TGF-β	Phy/TGF-β
**Sub G0**	24	1.03 ± 0.50	1.07 ± 0.22	2.33 ± 0.99	0.86 ± 0.12	1.46 ± 0.45	1.10 ± 0.07
	48	1.04 ± 0.25	0.94 ± 0.16	1.03 ± 0.40	2.54 ± 0.04	0.61 ± 0.23	1.82 ± 0.24
	72	0.77 ± 0.00	1.22 ± 0.07 *	1.94 ± 0.79	2.06 ± 0.35	1.14 ± 0.38	2.77 ± 0.94
**G1**	24	76.75 ± 3.88	74.55 ± 1.59	78.45 ± 4.37	73.25 ± 0.37	79.00 ± 0.33	73.45 ± 0.69
	48	87.10 ± 3.76	83.05 ± 2.74	87.20 ± 4.49	88.15 ± 0.53	81.80 ± 1.55	89.05 ± 1.18^Δ^
	72	87.60 ± 2.78	81.35 ± 0.37	84.90 ± 1.47	88.50 ± 1.14	79.40 ± 1.06	89.75 ± 0.86^Δ^
**S**	24	7.45 ± 2.90	11.15 ± 0.69	9.38 ± 0.84	12.05 ± 0.86	9.62 ± 0.72	12.20 ± 2.69
	48	4.72 ± 1.29	7.39 ± 0.88	6.87 ± 2.88	4.23 ± 0.48	8.10 ± 2.69	3.36 ± 0.27
	72	4.63 ± 1.23	6.61 ± 0.74	6.19 ± 1.27	5.45 ± 2.26	8.46 ± 0.19	3.14 ± 0.52
**G2/M**	24	14.80 ± 1.47	13.25 ± 2.49	9.84 ± 4.54	13.85 ± 0.61	9.95 ± 0.61	13.25 ± 2.08
	48	7.12 ± 2.22	8.63 ± 1.69	4.90 ± 1.20	5.06 ± 1.00	9.50 ± 0.90	5.78 ± 1.16
	72	7.00 ± 1.58	10.85 ± 0.45	6.95 ± 0.57	3.99 ± 0.80	11.00 ± 0.49	4.34 ± 1.27^Δ^

Values represent the mean ± standard deviation.(* *p* < 0.05 compared to control; Δ *p* < 0.05 compared to TGF-β).

## Data Availability

The data presented in this study are available in the Appendix A or can be provided by the authors.

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
