# Peer review of "Potential Effect of *Pseudevernia furfuracea* (L.) Zopf Extract and Metabolite Physodic Acid on Tumour Microenvironment Modulation in MCF-10A Cells"

_biomolecules, 2021, doi:10.3390/biom11030420_

Round 1

Reviewer 1 Report

Klaudia Petrova et al. provide flow cytometry, Western blot and immunofluorescence microscopy data regarding the modulation of TME (tumour microenvironment) and EMT epithelial-mesenchymal transition by lichens-derived compounds. The manuscript is of some interest, while deserving major revisions:

  1. While using flow cytometry, did the authors employed unstained, isotype controls? This should be highlighted.
  2. For all Western blot figures, densitometry readings/intensity ratio of each band should be included; the whole Western blot showing all bands and molecular weight markers should be included at least in the Supplementary Materials
  3.  Fluorescence microscopy: is the exposure time and normalization done while acquiring images in figure 6E?
  4.  Statistical analysis: a deeper explanation of the statistical methodology employed should be presented, while making clear statements regarding the tests used and the assumption made (normal distribution, parametric and non-parametric tests, etc.)
  5. I personally miss in introduction/discussion section some important insights that can increase the interest of the manuscript for a broad readership in the oncology field, especially focusing on breast cancer. In details, the tumor microenvironment (TME), defined as the complex ecosystem in which cancer cells interact with non-cancerous cells, represents an additional source of intra-tumor heterogeneity. The TME includes proliferating tumor cells, the tumor stroma, surrounding blood vessels, and immune cells. In particular, the dynamic interplay between cancer and immune cells has become an issue of great interest. There is growing recognition that immunoediting, the process whereby the immune system can both counteract and promote tumour development, contributes to cancer heterogeneity and represents a potential source of biomarkers (PMID: 32460897). At the same time TME function as a gatekeeper that allows cancer cell dissemination, also from cancer stem-cells standpoint: patients with epithelial cancers might present circulating cancer cells expressing mesenchymal rather than epithelial markers, because of epithelial-to-mesenchymal transition (EMT); the author should expand these topics, referring to PMID: 32460897. 

Author Response

Dear reviewer

we response to your valuable question in attached file.

Reviewer 2 Report

This study reports the results of tests of extracts of the lichen Pseudevernia furfuracea on tumour microenvironment modulation in MCF-10A cells. The authors conclude by noting that lichen extract regulated TME more strongly than physodic acid alone since there could have been a synergistic effect of the individual substances obtained in PSE.
The work is overall very thorough both from a methodological point of view and in its presentation and discussion. The methods for the extraction, purification and use of lichen substances in the tests are described in detail and correctly follow the standards normally used in the field. Similarly, the tests for potential anticancer effects are also entirely adequate.
The statistics used, although simple, effectively support the authors' conclusions.
I therefore have no particular comments to make on the article. it seems to me to be a really good piece of work. Perhaps a brief comment on the actual usability of these substances should be added at the discussion stage. Although lichens are organisms that are easily found in nature, their availability is all in all rather limited. In addition, the extraction yields of lichen substances are relatively low. The result is that, while working experimentally, the use of lichen substances on a larger scale could be quite complicated and could raise issues of conservation, economic yield of the drug product, and general applicability of the approach.

Author Response

(The authors gave the same response as above.)

Round 2

Reviewer 1 Report

The authors have clarified several of the questions I raised in my previous review. Most of the major problems have been addressed by this revision.